# The IVI Lab entry to the GENEA Challenge 2022 – A Tacotron2 Based Method for Co-Speech Gesture Generation With Locality-Constraint Attention Mechanism

Che-Jui Chang*
Sen Zhang*
chejui.chang@rutgers.edu
sen.z@rutgers.edu
Rutgers University
USA

Mubbasir Kapadia
Rutgers University
USA
mubbasir.kapadia@rutgers.edu

## ABSTRACT

This paper describes the IVI Lab entry to the GENEA Challenge 2022. We formulate the gesture generation problem as a sequence-to-sequence conversion task with text, audio, and speaker identity as inputs and the body motion as the output. We use the Tacotron2 architecture as our backbone with the locality-constraint attention mechanism that guides the decoder to learn the dependencies from the neighboring latent features. The collective evaluation released by GENEA Challenge 2022 indicates that our two entries (FSH and USK) for the full body and upper body tracks statistically outperform the audio-driven and text-driven baselines on both two subjective metrics. Remarkably, our full-body entry receives the highest speech appropriateness (60.5% matched) among all submitted entries. We also conduct an objective evaluation to compare our motion acceleration and jerk with two autoregressive baselines. The result indicates that the motion distribution of our generated gestures is much closer to the distribution of natural gestures.

## CCS CONCEPTS

• **Computing methodologies** → **Animation**; **Machine learning**; • **Human-centered computing** → *Virtual reality*.

## KEYWORDS

co-speech gesture generation, sequence-to-sequence modeling, locality constraint attention

**ACM Reference Format:**
Che-Jui Chang, Sen Zhang, and Mubbasir Kapadia. 2022. The IVI Lab entry to the GENEA Challenge 2022 – A Tacotron2 Based Method for Co-Speech Gesture Generation With Locality-Constraint Attention Mechanism. In *INTERNATIONAL CONFERENCE ON MULTIMODAL INTERACTION (ICMI '22), November 7–11, 2022, Bengaluru, India.* ACM, New York, NY, USA, 6 pages. https://doi.org/10.1145/3536221.3558060

*Both authors contributed equally to this research.

## 1 INTRODUCTION

Generating natural-looking and audio-matching gestures is a key component for embodied virtual agents as the body gesture is an important communication channel for human interlocution. Body gestures accompanied by speech help the interlocutors engage in the conversation and increase the perceived realism of an embodied conversational agent [15, 21]. Co-speech gesture generation aims to synthesize high-fidelity body motions that match the given audio or text semantics. It is generally believed that audio is the most informative input modality for gesture generation and several works in the literature [7] approach it using audio as the only input. Other prior works [3, 12, 22] try to address the problem by adding more modalities, including texts, speaker identities, and styles. In this work, we formulate the gesture generation problem as a sequence-to-sequence mapping with text, audio, and speaker identity as inputs and the joint representations as the output. Our proposed model utilizes the Tacotron2 architecture [6, 20, 25] as the backbone, where the sequence encoder and decoder are connected with the attention module. We concatenate the text and audio features as the input sequence and decode the body motions in an autoregressive manner. Tacotron2 was originally designed for speech synthesis and later used in voice transformation tasks [6, 20, 25], we propose several specific adjustments in the model architecture and the training paradigm for co-speech gesture generation. First, the locality-constraint attention mechanism is used in our model to limit the receptive field of a decoder token to its neighboring frames and forces the decoder to learn the local dependencies. Second, we add the velocity constraint for the optimization as it encourages the model to predict more expressive gestures. Third, we optimize the skeletal joint representations and the root positions for the full body gestures.

The collective evaluation results released by GENEA Challenge 2022 [24] indicate that our submitted entries for both tracks outperform the audio-driven and text-driven baseline methods in both human-likeness and speech appropriateness. Remarkably, our full body entry receives the highest appropriateness score among all submitted systems. Our code can be accessed via our repository[1]

## 2 RELATED WORK

Prior works [1, 3, 4, 8, 10, 12, 14, 22] for gesture generation usually take text, audio, or both as the inputs. The primary reason is the

---

[1]https://github.com/cjerry1243/Tacotron2-SpeechGesture

dependencies between gestures and these input modalities [7, 21]. From the perspective of an integrated framework for an embodied virtual agent [2, 15, 21], gesture synthesis is typically considered the final step with the textual response first predicted by the dialogue module and the audio synthesized from the predicted text. Generating gesture animations from the text and/or audio thus becomes a natural setup. In the following subsection, we review the methods in the literature for text-driven and audio-driven gesture generation. We also cover the multimodal gesture generation, where text, audio, and other inputs are used.

## 2.1 Gesture Generation from Text and Audio

According to [4, 12, 23], gestures are correlated with textual semantics. Methods for text-driven gesture generation aim to build a conversion model for the text-gesture correlation. For example, [4] applies a transformer-based network for affective gesture generation from texts and agent attributes. On the other hand, using only audio as the input is the most common approach for gesture generation in the literature. Audio provides more informative cues for generating natural-looking gestures. Some gesture features, such as path length, discovered by [7] strongly correlate with the audio signal. The works from [8, 10] use a convolution-based model with adversarial training for the hand and upper body gesture motions. Probabilistic generative models are also used in the literature for speech-driven gesture generation. StyleGestures [1] generates gestures from speech using normalizing flows. Audio2Gestures [14] applies a variational autoencoder for the audio-to-gesture conversion.

## 2.2 Multimodal Gesture Generation

Gesture generation from multiple input modalities enables the model to explore the missing dependency in one modality from another. Audio and text are commonly used as inputs. For instance, Gesticulator [12] concatenates audio features with word embeddings for autoregressive gesture prediction. Additional input modalities can also be used for the prediction. Trimodal [22] takes text, audio, and speaker identity as inputs and applies generative adversarial training for the optimization. Speech2AffectiveGestures [3] uses the same inputs and an additional seed pose sequence. Affective regulations are imposed on the seed sequence and the predicted gestures by the model for affect consistency.

Our proposed method uses text, audio, and speaker identity as inputs. We use a similar autoregressive manner and training paradigm for gesture generation as Gesticulator [12]. However, we observe that the autoregressive prediction by Gesticulator easily converges to average, less expressive gestures. We leverage the Tacotron2 architecture [20] as the backbone for the sequence-to-sequence prediction because its PreNet, attention, and PostNet structure help to generate more dynamic outputs.

## 3 CO-SPEECH GESTURE GENERATION

Our proposed system for co-speech gesture generation is based on Tacotron2 architecture [20]. Our model uses the same encoder and decoder structure as the backbone. We add the locality constraint [25] to the original attention module to strengthen the alignment of local audio and gesture features for the decoding. Our system

architecture is illustrated in Figure 1. We concatenate audio features, text features, and speaker identity as the input sequence. Our model would then predict the corresponding gesture sequence. We will introduce the critical operations used in the original Tacotron2 system and describe how specific adjustments are made for our co-speech gesture generation task in this section.

## 3.1 Tacotron2 Architecture

Tacotron2 was originally designed for text-to-speech conversion. Specifically, the input is the character or phoneme sequence, and the output is a sequence of mel-spectrogram. To describe the key operations in the system, we define $h = \{h_1, h_2, ..., h_t\}$ as the output of the encoder, where $t$ is the length of the input sequence. The attention context module would calculate the weight, $\alpha_i = \{\alpha_i^1, \alpha_i^2, ..., \alpha_i^t\}$, at the $i$-th decoding time step using the output of the attention LSTM at previous time step, $s_{i-1}$, the weight at previous time step, $\alpha_{i-1}$, and the encoder output $h$, as shown in Equation 1:

$$\alpha_i = \text{AttentionContext}(s_{i-1}, \alpha_{i-1}, h). \tag{1}$$

The attention vector $g_i$ is calculated by the linear combination of the attention weight $\alpha_i$ and $h$, as in Equation 2:

$$g_i = \alpha_i^T h. \tag{2}$$

The attention vector is then used in the attention LSTM and the decoder LSTM, as shown in Equations 3 and 4 respectively, for gesture generation.

$$s_i = \text{AttentionLSTM}(s_{i-1}, g_i, \text{PreNet}(y_i)). \tag{3}$$

$$d_i = \text{DecoderLSTM}(d_{i-1}, s_i, g_i). \tag{4}$$

## 3.2 Locality Constraint Attention

We add the locality constraint to the attention context because we want the decoder to learn the gesture-speech alignment from local audio features. The design of the locality constraint attention aligns with the results found in previous studies [16, 19] that the temporal correlation between gesture and speech is strong within a window of -1 ∼ 1 second. We achieve it by adding a mask, $m_i$, centered at the i-th decoding frame as an input to the attention context module, as shown in Equation 5. The resulting weight $\alpha_i$ still sums up to one while the value is zero at the time step where it is masked.

$$\alpha_i = \text{AttentionContext}(s_{i-1}, \alpha_{i-1}, h, m_i). \tag{5}$$

In practice, we use the left and right 30 frames (1 second) as the local window for attention.

## 3.3 Optimization

As the length of the output gesture sequence is the same as the length of the input sequence, we remove the stop token prediction that was present in the original model. Therefore, the remaining component in the loss function becomes:

$$\mathcal{L}_{taco}(y, \hat{y}, \hat{y}_{dec}) = \|y - \hat{y}\|_2 + \|y - \hat{y}_{dec}\|_2, \tag{6}$$

where $y$ is the ground truth, $\hat{y}$ is the predicted sequence by the model, and $\hat{y}_{dec}$ is the output sequence after the decoder. For optimizing the upper body gestures, we add a velocity constraint. The final loss function becomes:

$$\mathcal{L}_{upper} = \mathcal{L}_{joints} = \mathcal{L}_{taco}(y, \hat{y}, \hat{y}_{dec}) + \mathcal{L}_{taco}(\Delta y, \Delta \hat{y}, \Delta \hat{y}_{dec}), \tag{7}$$

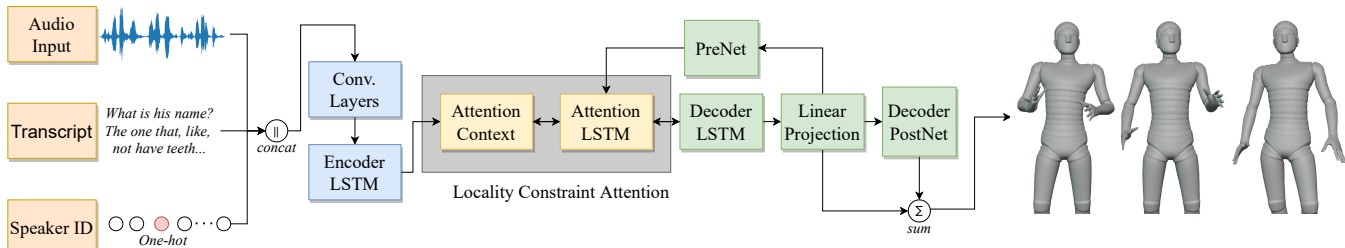

**Figure 1: Our proposed Tacotron2-based model with locality constraint attention mechanism (sec. 3.1 and 3.2). The architecture consists of an LSTM encoder, the locality constraint attention with the attention context and attention LSTM modules, and an autoregressive decoder. The audio features, text features, and speaker identity are concatenated as the model input for gesture generation.**

where the symbol, $\Delta$, means the motion velocity. For the full body gestures, the motion representation includes root positions. The range of the root positions is greater than the range of other joint representations, so we add a coefficient, $\lambda$, to its loss component, as shown in Equation 8.

$$\mathcal{L}_{full} = \mathcal{L}_{joints} + \lambda \mathcal{L}_{root}. \tag{8}$$

We use 0.01 for $\lambda$ for our FSH entry. Our model configuration is the same as the work from [25]. We use all training and validation data during the optimization. No external data is used. The autoregressive model is trained with teacher forcing. We use the Adam optimizer with a learning rate of 1e-4 and weight decay of 1e-6.

## 4  EXPERIMENT

### 4.1  Data Processing

We train our model using the data provided by GENEA Challenge 2022 [13, 24]. It includes a training set of 293 clips and a validation set of 40 clips. Each clip is about 1 minute long with audio, transcript, metadata, and gesture motion in BVH format. The sample rate of the audio is 44100 Hz. The motion FPS is 30. In this section, we detail the processing of each modality.

*Audio Processing.* For the audio features, we found that the mel-spectrogram was used in these prior works [1, 12] and the MFCC was used in [3] for affective gesture generation. We use both because we believe they are beneficial to gesture prediction. We also extract prosody features, including audio intensity, pitch, and their derivatives. Prosodies are important indicators for gesture generation. The audio energy is strongly related to emotional expressiveness, and pitch implies when hand gestures should pause.

All the three features, mel-spectrograms, MFCCs, and prosodies, are extracted at window length 4096 and hop length 1470, under the sample rate of 44100. The resulting audio features are 30 FPS, with the same sequence length as the motions. We use Librosa [18] package for the first two features. We set the number of filter banks to 64 for mel-spectrograms and use 40 dimensions for MFCCs. The prosodies are extracted using the Parselmouth [11] package. All audio features are normalized and concatenated before being passed to the model.

*Text Features.* The transcript includes words and their timings. To process the transcript, we use FastText word embeddings [5] with 300 dimensions. We leave all zeros for OOV words. We add two

additional dimensions to the text features, one labeling whether the corresponding audio frame is silent and the other telling if any laughter is present. The silence and laughter information can both be obtained from the transcript. We believe these two additional features are necessary because the silence information indicates when the gesture should start and pause, and the laughter information can be used by our model to infer specific laughing movements.

*Audio-Text Alignment.* We concatenate audio and text features by aligning them frame by frame. The timing of a word is matched to its corresponding audio frame. We observe that there may be more than one word in each timing slot, so we evenly divide the duration for those cases. We also add one-hot vectors for the 17 speakers. The resulting input dimension is 427.

**Table 1: The names of the joints we select as the gesture representations. The full body representation consists of all upper and lower body joints.**

|  | Joint Name |
|---|---|
| Upper Body | b_root, b_spine0, b_spine1, b_spine2, b_spine3, b_neck0, b_head, b_r_shoulder, b_r_arm, b_r_arm_twist, b_r_forearm, b_r_wrist_twist, b_r_wrist, b_l_shoulder, b_l_arm, b_l_arm_twist, b_l_forearm, b_l_wrist_twist, b_l_wrist |
| Lower Body | b_r_upleg, b_r_leg, b_r_foot, b_l_upleg, b_l_leg, b_l_foot |

*Motion Processing* We use the exponential map [9] as our motion representation. For the upper body, we select 19 joints. We add 6 lower body joints and root positions for the full body. The joint names are listed in Table 1. The finger motions are excluded because the finger data is not as accurate as other joints. Finally, the upper body pose has 57 dimensions, and the full body has 78 dimensions. For post-processing, we smooth the generated gestures using the Savitzky-Golay filter [17] with a window length of 9 and polynomial order of 3. We find the smoothing parameters filter out the jittering artifact and preserve better motion quality.

### 4.2  Evaluation Setup

The main evaluation of our method is done together with all other submitted entries by the organizers of the GENEA Challenge 2022

**Table 2: Summary of the collective perception study, with confidence level $\alpha$ = 0.05. "Percent matched" is the percentage of all matched responses and half equal responses. For the entry ID, the first letter "F" means full body while "U" means upper body. The second letter, "N" means natural gestures, "B" means baseline systems, and "S" means the submitted systems. Our two entries are FSH and USK.**

| ID | Human-likeness | | Appropriateness | | | |
| | Median | Mean | Number of responses | | | Percent matched |
| | | | Match. | Equal | Mismatch. | (splitting ties) |
|---|---|---|---|---|---|---|
| FNA | 70 $\in [69, 71]$ | $66.7 \pm 1.2$ | 590 | 138 | 163 | $74.0 \in [70.9, 76.9]$ |
| FBT | 27.5 $\in [25, 30]$ | $30.5 \pm 1.4$ | 278 | 362 | 250 | $51.6 \in [48.2, 55.0]$ |
| FSA | 71 $\in [70, 73]$ | $68.1 \pm 1.4$ | 393 | 216 | 269 | $57.1 \in [53.7, 60.4]$ |
| FSB | 30 $\in [28, 31]$ | $32.5 \pm 1.5$ | 397 | 163 | 330 | $53.8 \in [50.4, 57.1]$ |
| FSC | 53 $\in [51, 55]$ | $52.3 \pm 1.4$ | 347 | 237 | 295 | $53.0 \in [49.5, 56.3]$ |
| FSD | 34 $\in [32, 36]$ | $35.1 \pm 1.4$ | 329 | 256 | 302 | $51.5 \in [48.1, 54.9]$ |
| FSF | 38 $\in [35, 40]$ | $38.3 \pm 1.6$ | 388 | 130 | 359 | $51.7 \in [48.2, 55.1]$ |
| FSG | 38 $\in [35, 40]$ | $38.6 \pm 1.6$ | 406 | 184 | 319 | $54.8 \in [51.4, 58.1]$ |
| FSH | 36 $\in [33, 38]$ | $36.6 \pm 1.4$ | 445 | 166 | 262 | $60.5 \in [57.1, 63.8]$ |
| FSI | 46 $\in [45, 48]$ | $46.2 \pm 1.3$ | 403 | 178 | 312 | $55.1 \in [51.7, 58.4]$ |

(a) full body

| ID | Human-likeness | | Appropriateness | | | |
| | Median | Mean | Number of responses | | | Percent matched |
| | | | Match. | Equal | Mismatch. | (splitting ties) |
|---|---|---|---|---|---|---|
| UNA | 63 $\in [61, 65]$ | $59.9 \pm 1.3$ | 691 | 107 | 189 | $75.4 \in [72.5, 78.1]$ |
| UBA | 33 $\in [31, 34]$ | $34.6 \pm 1.4$ | 424 | 264 | 303 | $56.1 \in [52.9, 59.3]$ |
| UBT | 36 $\in [34, 39]$ | $37.0 \pm 1.4$ | 341 | 367 | 287 | $52.7 \in [49.5, 55.9]$ |
| USJ | 53 $\in [52, 55]$ | $53.6 \pm 1.3$ | 461 | 164 | 365 | $54.8 \in [51.6, 58.0]$ |
| USK | 41 $\in [40, 44]$ | $41.5 \pm 1.4$ | 454 | 185 | 353 | $55.1 \in [51.9, 58.3]$ |
| USL | 22 $\in [20, 25]$ | $27.2 \pm 1.3$ | 282 | 548 | 159 | $56.2 \in [53.0, 59.4]$ |
| USM | 41 $\in [40, 42]$ | $41.9 \pm 1.4$ | 503 | 175 | 328 | $58.7 \in [55.5, 61.8]$ |
| USN | 44 $\in [41, 45]$ | $44.2 \pm 1.4$ | 443 | 190 | 352 | $54.6 \in [51.4, 57.8]$ |
| USO | 48 $\in [47, 50]$ | $47.3 \pm 1.4$ | 439 | 209 | 335 | $55.3 \in [52.1, 58.5]$ |
| USP | 29.5 $\in [28, 31]$ | $32.4 \pm 1.4$ | 440 | 180 | 376 | $53.2 \in [50.0, 56.4]$ |
| USQ | 69 $\in [68, 70]$ | $67.5 \pm 1.2$ | 504 | 182 | 310 | $59.7 \in [56.6, 62.9]$ |

(b) upper body

[24]. The test set has the same format as the train and valid set, except that the 40 BVH files must be generated by the system and submitted as an entry to either full-body or upper-body track for collective evaluation. All the submitted BVH files are used to drive the motion of the same character without facial expressions and textures. Two subjective metrics, human-likeness and speech appropriateness, are reported in the collective perception study for both tracks. The score of human likeness represents how natural-looking the generated gesture is, and the speech appropriateness determines how well the generated gesture matches the input audio. The rating scale of human-likeness is 0-100 points. For appropriateness, two different gesture segments with the same audio, one from the correct input audio and the other randomly selected from other submitted files, are presented for user rating. The participants could choose either one as a better audio-matching gesture or select no preference. We kindly refer readers to the main challenge paper for more details regarding the experimental designs and evaluation setups.

## 5 DISCUSSION

### 5.1 Human-likeness

The evaluation results are shown in Table 2. Our submitted entries, the full body entry FSH and the upper body entry USK receive median human-likeness scores, 31 and 41, respectively. Among all submitted systems, our method generates fair motion quality. Statistically, our full body entry outperforms the other two entries, FSB and FBT, as described in Appendix A. Our upper body entry receives significantly better scores than four other entries, including the two baselines, UBT and UBA. However, there is a gap in human likeness between our entries and natural motion. We think one major reason is that our system does not include finger motions, which limits the upper bound of the human-likeness score our method can receive.

On the other hand, we conduct an objective evaluation to compare our method with two autoregressive baselines, Gesticulator [12], and the FBT entry [23]. Unlike the study conducted in the

**Table 3: Objective evaluation with the average jerk and acceleration. The mean and the standard deviation are shown in the table. The "small" represents the same model but with smaller model size. (We decrease the dimension of all hidden units to 256).**

|  | Average jerk | Average acceleration |
|---|---|---|
| Ours (FSH) | 3737.36 ± 842.43 | 131.78 ± 30.73 |
| Ours unsmoothed | 23401.66 ± 8608.51 | 478.14 ± 166.40 |
| Ours small unsmoothed | 15641.36 ± 4923.77 | 324.26 ± 97.32 |
| Gesticulator [12] | 2080.73 ± 256.68 | 68.38 ± 12.55 |
| FBT [23] | 1050.55 ± 301.54 | 49.19 ± 14.50 |
| Natural motion (validation) | 16888.87 ± 1950.56 | 373.04 ± 58.50 |
| Natural motion (FNA) | 16364.01 ± 1878.30 | 357.95 ± 54.14 |

challenge paper, we only calculate the metrics on the selected full-body joints because we would like to reduce the variance caused by finger motions. Table 3 reports the average jerk and acceleration. Our Tacotron2-based models obtain higher average jerk and acceleration than both baselines. The distribution of our generated motions is much closer to the distribution of natural motions. We also observe that the gestures generated by the baselines tend to be less expressive. They fail to move responsively to the prosody change in the audio. When comparing our method with natural motion, we find that the variances of our motion jerk and acceleration are relatively higher. For instance, the motion jerk deviation is greater than one-fifth of its mean, regardless of the smoothing or the mode size. The same ratio for natural motion is always below one-eighth. We believe the limitation of our proposed model is the stability of the generated gestures. Temporal smoothing must be applied after the model prediction, but the human-likeness score is therefore discounted.

## 5.2 Appropriateness

Regarding speech appropriateness, Table 2 reveals that our upper body entry gets 55.1% matched, and the full body entry obtains 60.5% matched. Remarkably, our full body entry receives the highest appropriateness score among all submitted systems, only second to the natural motion entry FNA. Statistically, our FSH entry is the only submitted entry that shows a significant difference when paired with four other systems, as mentioned in the challenge paper. Here we summarize several settings of our model that we think might be beneficial to improving speech appropriateness. First, the higher appropriateness score could possibly come from the design of the locality-constraint mechanism as the decoder is forced to consider its neighboring one-second window (-1 ∼ 1s) of features for gesture generation. While Gesticulator uses a similar window size (-0.5 ∼ 1s) to predict a frame of gesture based on the pioneer study [16, 19] for gesture-speech alignment, we believe our locality-constraint attention mechanism generalizes the concept of the alignment. It allows the model to weigh the local encoder features by itself when decoding. The second reason may be the usage of all the input features, including the mel-spectrogram, MFCC, prosody, and laughter. These multimodal inputs provide ample information for the model to learn to properly align speech with gestures, as our predicted gestures are pretty responsive to when the speaking starts and pauses. Nonetheless, our predicted gesture still falls behind the

natural motion by a margin. We hope our attempts to strengthening the local speech-gesture correlations in the architecture and the gestures generated by our system entries could facilitate further research on gesture generation in speech appropriateness.

## 6 CONCLUSION

We describe our system entry to the GENEA Challenge 2022 in this paper. We leverage Tacotron2 architecture as the backbone with the locality constraint attention mechanism that strengthens the local dependencies for gesture decoding. Our model optimizes the joint motions together with the root positions for the full body entry. The collective evaluation results indicate that our method outperforms the audio-driven and text-driven baselines in human likeness. We further conduct an objective evaluation to compare our method and two autoregressive baselines. Our generated gestures have higher average jerk and acceleration. We conclude that the distribution of our gesture motions is closer to the natural motions. Our full body entry receives the highest speech appropriateness score among all submitted entries. The potential reasons are raised in the discussion section.

When comparing our gestures with natural motion, we observe a gap in human likeness. Our autoregressive method cannot generate stable gesture movements with relatively high acceleration and motion jerk. In terms of speech appropriateness, we observe our generated gestures are responsive to when human voices start and pause. However, the natural gestures are still significantly more appropriate. We will focus on these aspects in our future work.

## 7 ACKNOWLEDGEMENT

The research was supported in part by NSF awards: IIS-1703883, IIS-1955404, IIS-1955365, RETTL-2119265, and EAGER-2122119. This material is based upon work supported by the U.S. Department of Homeland Security[2] under Grant Award Number 22STESE00001 01 01.

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

## A PAIRWISE SIGNIFICANT DIFFERENCE FOR HUMAN-LIKENESS

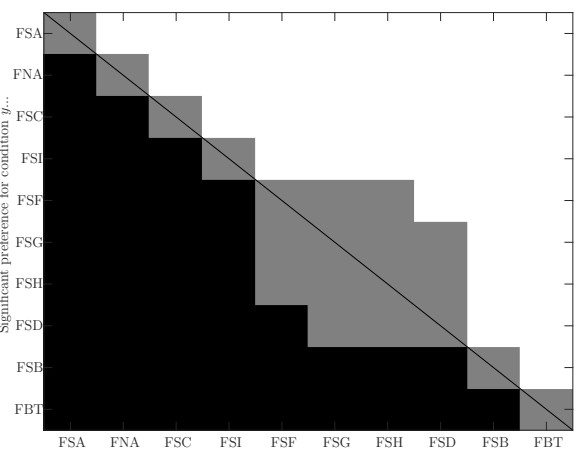

**(a) full body**

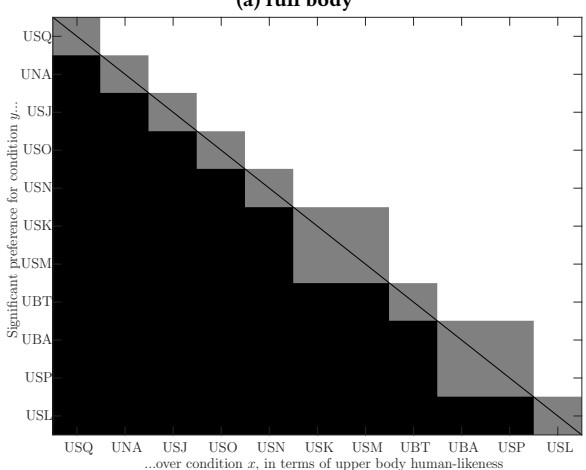

**(b) upper body**

**Figure 2: Significance of all pairwise entry differences. The white block means the entry at the y-axis is rated significantly above the entry at the x-axis, black represents the opposite, and the gray block means no significant difference at $\alpha = 0.05$.**

Figure 2 shows the pairwise significance study of all entries. Our full body entry, FSH, is rated significantly above FSB and FBT. The upper body entry, USK, is statistically above UBT, UBA, USP, and USL. However, our two entries are rated significantly below the natural motions, FNA and UNA.