# OpenReview forum: "The IVI Lab entry to the GENEA Challenge 2022 -- A Tacotron2 Based Method for Co-Speech Gesture Generation With Locality-Constraint Attention Mechanism"
_ACM.org/ICMI/2022/Workshop/GENEA — GENEA Challenge & Workshop 2022 Mainproceeding_

### Official Review · Reviewer_rZs5 · 2022-08-08
**There is not much novelty in the proposed approach and the results are also not significantly good.**

**Rating:** 5
**Confidence:** 4

**Review:**

This paper presents a Tacotron2-based method for co-speech gesture generation with locality constraint attention mechanism.
The architecture is based on the Tacotron2, which is largely used in TTS (text-to-speech) tasks.
In the proposed method, speech and subject id are added to the input along with the text information, in order to output the gesture sequences.
The methodology is straightforward and not so novel, but the application to the gesture generation task can be considered novel.
Average human-likeness below 50% have been achieved, indicating that the proposed approach was not enough to generate natural gesture motions.

No ablation studies are conducted.
No video samples should be provided (for example, through anonymous github links).

Eq.(6): it is not clear which decoder (LSTM or PostNet) is being referred.

---

### Official Review · Reviewer_rTfn · 2022-08-08
**A Tacotron2 Based Method for Co-Speech Gesture Generation With Locality-Constraint Attention Mechanism**

**Rating:** 8
**Confidence:** 5

**Review:**

# A Tacotron2 Based Method for Co-Speech Gesture Generation With Locality-Constraint Attention Mechanism

## Description

The paper describes a gesture generation system for the GENEA Challenge 2022. The method adapts Tacotron2, a model originally designed for text to speech synthesis.
The model is LSTM based with locally constrained attention. The model takes as input: audio features, word embeddings, and one-hot speaker embeddings. The output is exponential map rotations. The audio features are combined MFCC, mel-spectogram and prosodic features.

## References

The paper includes references to the most relevant recent work. I do not see the need to extend the references.

## Clarity of Exposition

The paper is generally well written, and the exposition was properly explained.
The authors include the evaluation for the participants of the GENEA challenge, and also, their own evaluation comparing jerk and acceleration.

The proposed method uses some modifications to Tacotron2. It would be interesting to include an ablation test to demonstrate the effect of the locality constraint.

During text processing, the authors add a label for laughter and silence. It would be interesting to see the ablation - particularly for laughter.

## Reproducibility

Even without releasing code, I am confident the work could be reproduced from the description.

## Conclusion

A well written paper that modifies an existing framework, Tacotron-2, in a logical way for this task. It would be interesting to see some ablation tests, but this is not necessary for inclusion.

The additional evaluation for jerk and acceleration is welcome, and could be extended to demonstrate the effectiveness of the modifications to the back-bone.

---

### Official Review · Reviewer_wmS3 · 2022-08-09
**An interesting approach with constrained attention and combined input features**

**Rating:** 7
**Confidence:** 3

**Review:**

Paper strengths:
- Overall, the paper is well written and easy to follow.
- The locality constraint attention is simple but efficient and well adapted to the task of gesture generation.
- The combined multiple audio features and additional silence/laughter labels help the model capture audio-gesture alignment.
- The proposed system receives the highest matched rate in the full-body tier, only second to the natural motion.

Comments and questions:
- What is the benefit of using both \hat{y} and \hat{y}_dec in the loss function? Also, it could be more helpful if \hat{y} and \hat{y}_dec are depicted in Fig. 1.
- The use of velocity constraints is a bit unclear. In line 173, why is the velocity constraint used only for upper body gestures? Does “upper body gestures” mean “upper body tier”? Or did the authors use masks for lower body joints during velocity loss calculation?
- How did the authors label the additional silence/laughter information? Since the authors state “these two additional features are necessary”, providing the details of the labeling process would be very helpful to readers.
- It would be better to include the box plots and bar plots of the evaluation results, if there is enough space. The plots help readers intuitively understand the results.

Minor points:
- The paper title should be “The YOUR_TEAM_NAME entry to the GENEA Challenge 2022”.
- Line 35: Reference [20] may be mismatched in the sentence “... [7, 20] approach it using audio as the only input”. The reference [20] is about a review of evaluation practices of gesture generation.
- Line 135: The second a_i^1 should be a_i^2.
- Audio sample rate: 44100 -> 44100 Hz

---

### Decision · Program_Chairs · 2022-08-11

**Decision:**

Accept (Main proceeding)

**Comment:**

Congratulations! A majority of reviewers recommended accepting this paper. Reviewers appreciated how well the paper is written. The chairs have decided to accept your paper to main ICMI track.

We suggest that the authors carefully consider the feedback received from the reviewers and use it to improve their manuscript for the challenge camera-ready submission deadline. Below follows some input from the chairs, based on the paper and the reviews:

How did the authors label the additional silence/laughter information? Since the authors state “these two additional features are necessary”, providing the details of the labelling process would be very helpful to readers.
Please include figures and box plots to make it more pleasant for readers to interpret the results.